# Adsorption Capacity of Silica SBA-15 and Titanosilicate ETS-10 toward Indium Ions

**DOI:** 10.3390/ma16083201

**Published:** 2023-04-18

**Authors:** Inga Zinicovscaia, Nikita Yushin, Doina Humelnicu, Dmitrii Grozdov, Maria Ignat, Ionel Humelnicu

**Affiliations:** 1Department of Nuclear Physics, Joint Institute for Nuclear Research, Joliot-Curie Str., 6, 1419890 Dubna, Russia; 2Department of Nuclear Physics, Horia Hulubei National Institute for R&D in Physics and Nuclear Engineering, 30 Reactorului Str. MG-6, 077125 Magurele, Romania; 3Faculty of Chemistry, Alexandru Ioan Cuza University of Iasi, Bld. Carol I, 11, 700506 Iasi, Romania

**Keywords:** adsorption, indium, silica SBA-15, titanosilicate ETS-10, pollution, remediation

## Abstract

Indium is an extremely important element for industry that is distributed in the Earth’s crust at very low concentrations. The recovery of indium by silica SBA-15 and titanosilicate ETS-10 was investigated at different pH levels, temperatures, times of contact and indium concentrations. A maximum removal of indium by ETS-10 was achieved at pH 3.0, while by SBA-15 it was within the pH range of 5.0–6.0. By studying kinetics, the applicability of the Elovich model for the description of indium adsorption on silica SBA-15 was shown, while its sorption on titanosilicate ETS-10 fitted well with the pseudo-first-order model. Langmuir and Freundlich adsorption isotherms were used to explain the equanimity of the sorption process. The Langmuir model showed its applicability for the explanation of the equilibrium data obtained for both sorbents, the maximum sorption capacity obtained using the model constituted 366 mg/g for titanosilicate ETS-10 at pH 3.0, temperature 22 °C and contact time 60 min, and 2036 mg/g for silica SBA-15 at pH 6.0, temperature 22 °C and contact time 60 min. Indium recovery was not dependent on the temperature and the sorption process was spontaneous in nature. The interactions between the indium sulfate structure and surfaces of adsorbents were investigated theoretically using the ORCA quantum chemistry program package. The spent SBA-15 and ETS-10 could be easily regenerated by using 0.01 M HCl and reused with up to 6 cycles of adsorption/desorption with a decrease in the removal efficiency between 4% and 10% for SBA-15 and 5% and 10% for ETS-10, respectively.

## 1. Introduction

Indium is an important metal, being one of the least abundant elements dispersed in the Earth’s crust. Indium in nature is present accompanying zinc, iron and copper in minerals and prior to its use, it has to be recovered from these minerals. Indium has a poor distribution on the Earth (about 0.05 ppm in terrestrial resources and 0.072 in oceanic crust), together with its presence just in a several world regions, thus, there will be a serious deficiency in its supply in the future. Indium, especially as indium–tin oxide (ITO), is widely used in the manufacturing of liquid crystal displays (LCDs) and plasma displays, semiconductor devices, in photovoltaic solar panels and infrared photodetectors [1,2,3].

Recently, the presence of indium in industrial effluents and its possible release in water bodies is of significant interest due to possible adverse environmental consequences [4,5]. Indium has no metabolic role for animals and humans, but can be toxic and harmful for the kidney, heart, liver, bone marrow, immune system and lungs [6,7].

In recent years, the necessity for indium has increased intensely due the development of the optoelectronic and semiconductor materials industry. Due to the fact that indium demands are increasing permanently and due to its toxicity to living organisms, new techniques for the recovery of indium(III) ions from industrial wastewaters have been developed [8,9,10,11,12].

The most widely used conventional methods for indium ion recovery from industrial wastewaters refer to electroanalytical techniques [13,14], solvent extraction [15,16], co-precipitation [17], biosorption [18] and liquid membrane separation [19]. Nevertheless, these methods have serious disadvantages, including the incomplete recovery of the indium from aqueous solutions, high demand of reagents (especially organic solvents), high energy consumption, significant capital cost and generation of other waste products.

In recent years, adsorption became one of the most appropriate approaches for indium recovery because of its many advantages, such as a simplicity and safety of the process, low cost, environmental friendliness, the ease of adsorbents regeneration and high recovery efficiency [20,21,22].

Several research studies were performed with the purpose of finding a suitable adsorbent material with superior selectivity to separate and recover indium ions from wastewaters. All the following adsorbents were tested: starch, activated carbon [23], fly ash, modified chitin [2], spent coffee grounds [9], nanomaterials [14], peanut shell pellets, clay minerals, zeolites and polymeric resins that have grafted different functional groups [20,21,24]. In most of the adsorption processes, the indium ion removal was possible due to metal ions chelating to the functional groups on the surface of the adsorbent materials. Thus, Calagui et al. [22] investigated the adsorption of indium ions from an aqueous solution using chitosan-coated bentonite beads as the adsorbent. Moreover, Zhao et al. [25] investigated the removal of indium ions from different simulated wastewaters by means of magnetic cobalt ferrite (CoFe_2_O_4_)-coated zeolite.

As is well known, voids at the nanometric scale have been recognized to play an important role in adsorption due to a phenomenon assigned to the electrical double layer (EDL) at the solid–water interface that increases in compression as the pore size decreases [26]. Actually, this phenomenon occurs in confined spaces (pores) where the well-known fundamental properties of water molecules are changed, affecting in this way the adsorption of cations at solid–water interfaces [27]. For this reason, nowadays many researchers focus on the development of micro-mesoporous materials (with pore diameter ≤ 50 nm) with various chemical compositions.

Among the different mesoporous materials, SBA-15 has some properties including: a uniform ordered structure containing cylindrical and parallel mesopores, high surface area, high thermal and mechanical stability, regular and adjustable pore size, inertness and is harmless to the environment. Due to these advantages, ETS-10 can be regarded as a promising adsorbent for the removal of metallic ions from residual waters. The maximum adsorption capacity of SBA-15 for Cu(II) was 11.39 mg/g [28]. Da’na and Sayari applied with good results the amine-functionalized SBA-15 to real water in order to recover Cd(II), Co(II), Cu(II), Zn(II), Pb(II), Ni(II), Al(III) and Cr(III) [29]. The maximum adsorption capacity of a magnetic SBA-15 nanosorbent was 140 mg/g for Cd(II), 122 mg/g for Ni(II), 110 mg/g for Pb(II) and 115 mg/g for Zn(II), respectively [30].

On the other hand, ETS-10 is a microporous material with a particular architecture (a three dimensional 12-ring pore system), which has a high thermal stability and can be easily regenerated. These properties recommend its use in the removal of pollutants from wastewaters. Thakkar et al. [31] demonstrated that rare-earth elements can be recovered by using ETS-10 with a maximum adsorption capacity of 0.5491 mmol/g for Y(III), 0.4723 mmol/g for Nd(III), 0.6519 mmol/g for Eu(III), 0.7206 mmol/g for Tb(III), 0.7549 mmol/g for Dy(III) and 0.5731 mmol/g for Ni(II), respectively.

Considering the above results, the main objectives of the present study are: (i) to investigate the effect of initial In(III) concentration, pH, time and temperature on indium ion recovery using silica SBA-15 and titanosilicate ETS-10 sorbents; (ii) to describe the kinetics, equilibrium and thermodynamics of the adsorption process and; (iii) to propose a theoretical mechanism model to describe indium recovery by silica SBA-15 and titanosilicate ETS-10.

## 2. Materials and Methods

### 2.1. Silica SBA-15 and Titanosilicate ETS-10 Preparation

The amphiphilic triblock copolymer poly(ethylene glycol)-block-poly(propylene glycol)-block-poly(ethylene glycol) (Pluronic P123 EO_20_PO_70_EO_20_; Sigma Aldrich, Darmstadt, Germany) was used as a soft template in the synthesis of SBA-15 silica. For the SBA-15 synthesis, the mixture containing Pluronic P123 in the amount of 4 g was dissolved in 150 mL of HCl (2 M) solution and stirred until the formation of surfactant micelles. Next, TEOS (tetraethyl orthosilicate, Sigma-Aldrich) as a silica source in the amount of 9 g was added dropwise to the prepared mixture under continuous stirring. The resulting gel was aged at 40 °C for 24 h and again subsequently at 100 °C for 48 h, with continued stirring for the entire time. The obtained white precipitate was separated by filtration, washed several times with distilled water, and dried firstly overnight in an oven at 100 °C, and then calcinated for 6 h in a muffle furnace at 550 °C with a heating rate of 1 °C/min. [32].

In a typical synthesis of titanosilicate ETS-10 (composition 3.4Na_2_O:1.5K_2_O:TiO_2_:5.5SiO_2_:150H_2_O), NaCl (Chemical, Iasi, Romania) in the amount of 1.62 g and KCl (Chemical, Romania) in the amount of 2.48 g were dissolved in ultrapure water. Next, sodium silicate solution containing sodium metasilicate Na_2_SiO_3_ (reagent grade, Chemical, Iasi, Romania) in the amount of 9.41 mL was added dropwise to the mixture. Thus, the Si precursor solution (solution A) was obtained. Further, to prepare the Ti source suspension (solution B), 0.88 g of TiO_2_ (Degussa-P25, Sigma-Aldrich) were suspended in 7.8 mL of H_2_SO_4_ (9%), resulting solution B. Thereupon, under vigorous hand-shaking the obtained solutions, A and B were mixed, resulting in a viscous mixture that was hand-shaken for another 5 min. The obtained gel was transferred into a Teflon-lined stainless-steel autoclave and kept for 72 h at 230 °C in a low heat oven. Next, the reaction mixture was cooled at room temperature, centrifuged, washed several times with distilled water, and dried overnight at 50 °C [32].

All chemicals used to obtain the adsorbents as well as those in the adsorption experiments were of analytical grade.

### 2.2. Adsorption Experiments

To prepare solutions with the desired indium concentrations, In_2_(SO_4_)_3_ obtained from Sigma-Aldrich (Darmstadt, Germany) was used. The mass of the sorbent in the performed experiments was 20 mg and the volume of the experimental solutions was 10 mL, and the indium(III) concentration was 10 mg/L (except in the equilibrium experiments). The effect of different initial pH values was studied in the range of 2.0–6.0 with an initial indium concentration of 10 mg/L at 22 °C. To obtain the required pH of the solution, 0.1 M NaOH or HCl were used. The effect of other parameters on indium adsorption was studied at optimal pH values. The effect of the initial indium concentration was investigated in the range of 10–100 mg/L at 22 °C. The kinetics of the sorption was measured by varying the initial time of contact from 1 to 60 min, while the thermodynamics was evaluated in the temperature range of 20–50 °C. Adsorption experiments were performed on a rotary Unimax 1010 shaker (Heidolph, Schwabach, Germany) at a fixed agitation speed of 200 rpm. At the end of the experiments, the sorbents were separated from the solution by filtration. All experiments were performed in triplicate.

The amount of indium adsorbed per unit of used sorbents and the efficiency of indium ion removal were calculated using Equations (1) and (2):(1)$$q=\frac{V\left(C_{i}-C_{f}\right)}{m}$$
(2)$$R=\frac{C_{i}-C_{f}}{C_{i}}\times 100$$
where *q* is the adsorption capacity, mg/g; *V* is the volume of solution, L; *C_i_* and *C_f_* are initial and final concentrations of indium in mg/L; and *m* is the weight of sorbent, g.

### 2.3. Desorption Studies

In order to prove the multiple use of the studied adsorbents, titanosilicate ETS-10 and silica SBA-15, the desorption of In(III) ions and re-use of the adsorbents in several adsorption/desorption cycles was carried out. The adsorbents loaded with In(III) were eluted with a 0.01 M HCl aqueous solution. Then, after washing several times with distilled water, the adsorbents were reused in another adsorption cycle.

### 2.4. Methods

Indium concentrations in the initial solutions and after the batch experiments were measured using ICP-OES PlasmaQuant PQ 9000 Elite spectrometer (Analytik Jena, Jena, Germany). The adsorbents’ surfaces were described applying the S3400N Scanning Electron Microscope (Hitachi, Waltham, MA, USA) supplied with an energy-dispersive X-ray spectroscope. Infrared spectra were obtained using a Bruker Alpha Platinum-ATR spectrometer (Bruker Optics, Ettingen, Germany). XRD measurements of the adsorbents before and after indium(III) removal were performed with the Empyrean Multi-Purpose Research X-Ray Diffractometer. The Cu anode (CuKα: λ = 1.541874 Å) was used as a source of radiation.

### 2.5. Data Analysis

The kinetics of the indium sorption was described using three kinetic models (Equations (3)–(5)).

The pseudo-first-order model (PFO):(3)$$q_{t}=q_{e\;}\left(1-e^{-k_{1}t}\right)$$

The pseudo-second-order model (PSO):(4)$$q=\frac{q_{e}^{2}k_{2}t}{1+q_{e}k_{2}t}$$

The Elovich model (EM):(5)$$q_{t\;}=\frac{1}{\beta }ln\left(1+\alpha \beta t\right)$$
where *q_t_* is the amount of indium adsorbed (mg/g) at time t, (mg/g); *k*_1_ (1/min) and *k*_2_ (g/mg·min) are the pseudo-first-order and the second-order reaction rate equilibrium constants, *α* (g/mg∙min); and *β* (g/mg) are Elovich model constants.

Equilibrium data were described using Langmuir and Freundlich isotherm models (Equations (6) and (7)).
(6)$$q_{m}=\frac{q_{m\;}bC_{e}}{1+bC_{e}}$$
(7)$$q_{m}=K_{F}Ce^{\frac{1}{n}}$$
where *C_e_* is the metal concentration at equilibrium (mg/L); *q_m_* is the maximum adsorption capacity (mg/g), *b* (L/mg); and *K_F_* and *n* are Langmuir and Freundlich equation constants.

The separation factor *R_L_* was calculated by Equation (8).
(8)$$R_{L}=\frac{1}{1+bC_{i}}$$

An *R_L_* value less than one unit means that adsorption is favorable and *R_L_* values higher than one unit means that adsorption is unfavorable.

The thermodynamic parameters such as the standard free energy (Δ*G*^0^), enthalpy change (Δ*H*^0^) and entropy change (Δ*S*^0^) were estimated from Equations (9) and (10):(9)$$\ln K_{d}=\frac{\Delta S^{0}}{R}-\frac{\Delta H^{0}}{RT}$$
(10)$$\Delta G^{0}=\Delta H^{0}-T\Delta S^{0}$$

The distribution coefficient *K_d_* was calculated by Equation (11):(11)$$K_{d}=\frac{C_{a}}{C_{e}}$$
where *C_a_* is the concentration of indium(III) adsorbed, mg/L.

## 3. Results and Discussion

### 3.1. Adsorbents’ Characterization

The SEM images of the silica SBA-15 and titanosilicate ETS-10 are presented in Figure 1. In the case of silica SBA-15, an agglomeration of rod-shaped particles, characterized by a mesoporosity, with the size of 300–500 nm was seen (Figure 1a). The titanosilicate ETS-10 sorbents consisted of quasi-cubic crystals forming agglomerations (Figure 1b).

The energy-dispersive X-ray spectroscopy analysis showed the presence of Si and O in silica SBA-15, indicating its chemical purity. Ti, Si, Na and K were present in the spectrum obtained for titanosilicate ETS-10. The Na^+^/K^+^ ratio of 2.8 calculated using the EDX results for titanosilicate ETS-10 was close to the value presented in [33]. The BET surface area of silica SBA-15 was 803 m^2^/g, exhibiting cylindrical pores with the mean pore size of 7–8 nm. The BET surface area of titanosilicate ETS-10 was significantly lower and constituted 31 m^2^/g with a mean pore size of 0.8 nm [32]. The volume of the pores for silica SBA-15 was 1.11 cm^3^/g and for titanosilicate ETS-10 it was 0.0438 cm^3^/g.

The XRD patterns of the adsorbents before and after indium(III) sorption are presented in Figure 2. The EXD measurements showed the amorphous structure of silica SBA-15 adsorbent. In the XRD pattern of titanosilicate ETS-10, both board and narrow reflections were presented, indicating a high crystalline but disordered material [34]. All peaks observed in the spectrum correspond to those reported previously for ETS-10 [35,36]. An intense peak at 2θ = 25 reports for the crystalline state of ETS-10 microporous titanosilicate [37]. Moreover, diffraction patterns that reveal anatase and rutile crystalline phases were observed. Indium(III) adsorption did not modify the structure of the adsorbents.

The FTIR spectra of the analyzed sorbents are shown in Figure 3. In the spectrum of silica SBA-15 (Figure 3a), the bending vibration and symmetric stretching bond of Si–O–Si were seen at 500 cm^−1^ and 800 cm^−1^, respectively. The band at 950 cm^−1^ was attributed to the Si-OH group and at 1000 cm^−1^ to the asymmetric stretching vibration of Si–O–Si. The band at 3400 cm^−1^ could be attributed to water molecules. The bands at 3000–2850 cm^−1^ and 1470 cm^−1^ are characteristic for C–H groups and at 1720 cm^−1^ for the carboxylic group, C=O. The adsorption of indium ions resulted in the decrease in the intensity of the functional groups, except Si–O–Si, indicating their involvement in indium ion adsorption.

In the titanosilicate ETS-10 (Figure 3b), the spectrum peaks in the region of 3000–2850 cm^−1^ could be attributed to OH and NH_2_ groups and at 1645 cm^−1^ to the bending vibration of water molecules. The adsorption band at 1420 cm^−1^ is characteristic for stretching vibrations of the C–O group. A peak at 1380 cm^−1^ can be assigned to C–H bending, while bands in the region of 1300–850 cm^−1^ belong to the asymmetric stretching vibration of Si–O–Si and Si–O–Ti bonds. Bands at 943 cm^−1^ and 880 cm^−1^ could be characteristic of Si–O− and Ti–O− terminal bonds forming. In the spectrum of the sorbent obtained after indium ion adsorption, changes in the intensities of the peaks in the area of OH groups after adsorption and/or ion exchange processes were observed. The detailed characteristics of the analyzed sorbents is presented in [32].

### 3.2. The Effect of Experimental Parameters on Indium Ion Adsorption

The effect of pH on the indium adsorption was studied in the pH range of 2.0–6.0. Experiments were not performed at higher pH values due to the formation of insoluble indium hydroxide that makes the adsorption process more difficult. As can be seen in Figure 4a, the lowest adsorption of indium constituting 17% on both sorbents was observed at pH 2.0, showing that a very acidic pH is unfavorable for indium ion removal. At low pH values, functional groups are tightly associated with H_3_O^+^ that restricts the approach of metal cations because of repulsive forces [38,39,40].

In the case of titanosilicate ETS-10, the highest removal of indium (99%) was attained at pH 3.0, then electrostatic interactions between indium ions and the negatively charged functional groups seemed to be responsible for the metal ions sorption [40]. A further increase in the pH resulted in the drastic decrease in indium removal up to 16% at pH 5.0. Indium mainly exists as In^3+^, In(OH)^2+^ and In(OH)^2+^ at a pH below 3.0, and at pH > 3.0 In(OH)_3_ precipitate begins to form [41]. Since NaOH was used to adjust the pH, a decrease in indium(III) adsorption can be explained by its precipitation as well competition with OH groups for binding sites [39]. A further increase in the pH resulted in the drastic decrease in indium removal up to 16% at pH 5.0, which can be attributed to a reduction in the concentration of positively charged groups on the sorbent’s surface [39]. By this decrease, it could also be assumed that indium ions predominantly bound to HO^-^ groups. However, at pH 6.0 the titanosilicate ETS-10 removal capacity toward indium increased by up to 74%. This behavior can be related to disparities in the sorbent or indium precipitation [39].

The silica SBA-15 behaved differently compared to titanosilicate ETS-10. The sorbent removal capacity increased from 85% at pH 3.0 to 99% at pH 6.0. According to [28], the nanomesoporous SBA-15 molecular sieve has the optimum adsorption effect for metal ions under weak acidic conditions (pH 3.0–4.0). Thus, indium ion removal on the level of 85–87% at pH 3.0–4.0 can be associated with metal ions binding to functional groups. The high indium removal at higher pH values could be explained by its precipitation, due to the use of NaOH for the increasing pH values and the consequent growth in the concentration of OH groups in the solution. This finding is in agreement with [22], who showed that at a higher pH, the removal of indium occurred due to both the adsorption and precipitation of In(OH)_3_.

In contrast to the present study, Alguacil et al. [23] reported maximum indium removal at pH 10.0. Indium sorption on UiO-66 [41], Shewanella algae [18] and brown algae *Ascophyllum Nodosum* [42] were performed at pH 3.0.

The effect of time on indium ion removal is shown in Figure 4b. Adsorption of indium ions onto titanosilicate ETS-10 was a very quick process. In 3 min of sorbent–sorbate interaction, 98% of the indium ions were removed from the solution and then an equilibrium was established. The fast phase of indium sorption is explained by external surface adsorption, which occurs instantaneously. The second slow stage is the gradual adsorption stage before the equilibrium is reached [43]. For silica SBA-15, the equilibrium was attained after 45 min of interaction, when 97% of the indium ions were removed from the solution. It should be mentioned that in the first minute of the interaction, 62% of the indium ions were removed from the solution. Fast sorption of indium in the first minutes of interaction is possible due to a larger number of functional groups on the adsorbent surface, while the lowering of indium removal can be explained by the exhaustion of the adsorption sites [38].

The effect of the initial indium concentration on its adsorption on studied sorbents is shown in Figure 4c. An increase in the indium concentration in the solution resulted in the growth of the sorbents’ adsorption capacity from 8 to 81.5 mg/g for silica SBA-15 and from 8.6 to 81.6 mg/g for titanosilicate ETS-10. Usually, an increase in the metal concentration is associated with the decrease in the metal removal capacity [38,44]. However, in the present study the efficiency of indium ion removal by both sorbents was maintained on the level of 97–99% regardless of the metal concentration in the solution.

The influence of temperature on indium removal is presented in Figure 4d. An increase in the temperature from 20 to 50 °C did not significantly influence the removal of indium by the studied sorbents, as it was on the level of 96–99%. This suggests that the indium(III) adsorption process is not strictly related to its endothermic nature and is a physical adsorption [45]. The adsorption of indium ions onto chitosan-coated bentonite beads [22] and nanotubes [23] was favorable at a high temperature.

### 3.3. Kinetics, Equilibrium and Thermodynamics of the Indium Adsorption

Indium(III) adsorption onto silica SBA-15 and titanosilicate ETS-10 was interpreted in terms of kinetics by the use of the pseudo-first order, pseudo-second order and Elovich kinetic models, which in their non-linear forms are illustrated in Figure 5 and the parameters calculated from the models are given in Table 1.

According to coefficients of correlation values, the sorption of indium ions on silica SBA-15 was well described by the Elovich model. The Elovich model is used to describe adsorption onto heterogeneous surfaces, pointing to the fact that chemisorption is one of the main mechanisms of metal sorption [46].

The adsorption of metal ions onto mesoporous adsorbents can be presented as a two-step process. In the first stage, indium ions move to the boundary layer surrounding the mesoporous matrices and in the second stage they are trapped to functional groups on the sorbent surface [32]. Chemisorption was the rate-determining step of the adsorption of indium(III) using chitosan-coated bentonite beads [22].

Considering the coefficient of determination values, it can be concluded that all the applied models can be suitable to describe indium sorption onto titanosilicate ETS-10. However, extremely high values of α in the Elovich model indicate its inconsistency for the description of experimentally obtained data. The close agreement between experimental and calculated adsorption capacities supported the applicability of the pseudo-first- and pseudo-second-order models for the description of indium ion removal. To determine the goodness of fit of the kinetic models, the Akaike information criterion (AIC) test was applied, which showed that the pseudo-first-order model was more suitable for the explanation of the experimental data. The model consider that the rate of adsorption sites occupation is proportional to the number of unoccupied sites [37]. This suggests that the rate-limiting step of indium ion sorption onto the titanosilicate ETS-10 adsorbent is dependent on the concentration of the indium ions in the solution [47]. Thus, indium sorption on titanosilicate ETS-10 was a conventional mass transport phenomenon rather than chemical adsorption [48].

The Langmuir and Freundlich adsorption isotherms were explored to explain the steadiness of the indium adsorption process. The plots of the models are presented in Figure 6 and parameters are given in Table 1.

Based on the determined coefficients values, it may be concluded that both equations fit the experimental data reasonably well. However, the AIC test showed that the Langmuir model, which assumes a monolayer adsorption onto the surface of an adsorbent with a finite number of identical sorption sites, is more correct. Even the values of the experimental adsorption capacity for both sorbents were very close to the calculated capacity for silica SBA-15, which was 5.5 times higher than for titanosilicate ETS-10 sorbent. The higher adsorption capacity of silica SBA-15 was also supported by the greater Langmuir adsorption constant (b) value, which indicates the affinity of SBA-15 to indium ions. The R_L_ values of 0.98 for silica SBA-15 and 0.85 for titanosilicate ETS-10 were below 1.0, proving that adsorption was a favorable process.

The values of the Freundlich constant K_F_, which relates to the adsorption capacity, for titanosilicate ETS-10 were higher than for silica SBA-15, indicating that the loading indium was higher on this sorbent. The values of 1/n 0.98 for silica SBA-15 and 0.77 for titanosilicate ETS-10 showed good adsorption of indium ions onto heterogeneous surfaces [49]. The good fit of the experimental data to the Langmuir isotherm indicates a more physical sorption mechanism [41].

The adsorption of indium ions onto chitosan-coated bentonite beads is best described by the Langmuir model with a maximum sorption capacity of 17.8 mg/g. Indium adsorption on poly(vinylphosphonic acid-co-methacrylic acid) microbeads corresponded better to the Langmuir model with q_m_ in the range 0.47–0.70 mmol/g [20]. The Langmuir isotherm fitted the experimental data better than the Freundlich isotherm for modified solvent impregnated resins, with q_m_ in the range of 26–59 mg/g [21]. The best adsorption performance of the waste and natural biomass of *Ascophyllum nodosum* for indium was between 48 and 63 mg/g [42], while of UiO-66 sorbent was 11.8 mg/g [41]. The maximum sorption capacity obtained for sorbents studied in the present work is significant if compared to other indium sorbents.

The efficiency of the two investigated adsorbents, ETS-10 and SBA-15, for the removal of In(III) was compared with the maximum adsorption capacity presented in the literature for other adsorbents (Table 2).

As can be seen, high sorption capacities were reported for some adsorbents, such as: raw chitin [2], supercritical-modified chitin [2] and SBA-15 (present study), respectively. However, the lowest adsorption capacity was reported for CoFe_2_O_4_-zeolite [25], UiO-66 [37] and chitosan-coated bentonite beads, respectively.

ΔH° and ΔS° parameters were obtained from the plot of lnK_d_ against 1/T (Appendix A). The calculated values are summarized in Table 3.

The negative ΔG° values indicate the spontaneous nature of the indium adsorption process. Since the ΔG° values were between −20 KJ/mol and 0 KJ/mol, it can be suggested as a physisorption process [50]. The positive value of ΔH° indicates an endothermic character of indium sorption onto silica SBA-15, while its sorption onto titanosilicate ETS-10 was exothermic. A positive value of ΔS° for silica SBA-15 indicates an increase in randomness at the solid/solution interface during adsorption [44]. For titanosilicate ETS-10 sorbent, the negative ΔS° value was obtained, pointing at the decrease in the randomness of the system during adsorption [51]. The positive values of ΔS° for silica SBA-15 indicates the high affinity of the adsorbent for the indium(III). This fact was also confirmed by a higher maximum adsorption capacity value calculated with the Langmuir model.

### 3.4. Theoretical Investigations

The maximum adsorption capacity calculated from the Langmuir model was 366 mg/g for titanosilicate ETS-10 and 2036 mg/g for silica SBA-15. This difference could be associated with the available specific surface area of the investigated materials on which the adsorption of indium ions occurred. The SBA-15 silica had the specific surface area (803 m^2^/g) of about 26 times higher than that of ETS-10 titanosilicate (31 m^2^/g), thus providing a larger surface and more active sites for the physisorption of indium ions.

The interactions between the indium sulfate structure and adsorption surfaces was also theoretically investigated using the ORCA quantum chemistry program package [52,53]. The hybrid functional B3LYP [54,55] with double-zeta basis sets def2 [56] and Density Functional Theory (DFT) theory [57] were employed for the study of the conformational indium sulfate structure in the gas phase and in a water solution. A small fragment, according to the crystalline structures from the Crystallography Open Database (COD), was used for the simulation of the adsorbent macrostructures. For the molecular modeling, the pentahydrate with the five water molecules form [58] for the indium sulfate was considered. The ideal spatial structure of pentahydrate indium(III) sulfate cation, considered as the start structure in our theoretical investigations, with an octahedral configuration for In and 3.2 Å between In and S and 2.16 Å for the In–w distance, respectively [59], is illustrated in Figure 7a. Two hydrogen-bond-type interactions can be observed between two water molecules from the equatorial configuration of the indium atom with two oxygen atoms from the sulfate group. Due to these interactions, the two protons involved passing from the water molecule at the sulfate group forming a sulfuric acid molecule, and the indium atom will have two covalent bonds with the hydroxyl group (2.03 Å for In-OH and 2.2 Å for In–w), according to Figure 7b. If the environment in which the investigated structure is located is characterized by a high pH value, then indium passes into the hydroxylated form, In(OH)_3_, and separates as a precipitate. This latter structural configuration of indium can be encountered in an interaction with an edge or peak on the adsorbent surface, because when approaching the surface, the crystallization of the water will be gradually removed and may leave the area of interest or may be found around the active center by establishing hydrogen bonds.

Near to the surface of the ETS-10 adsorbent, a water molecule is removed from the coordination sphere of the indium atom and a covalent bond is formed with an oxygen atom on the surface, according to Figure 8a. Next, the indium structure can form another bond with an oxygen atom keeping the octahedral structure, with two water molecules in the axial plane, with two bonds In–O, In–OSO_3_ and In–w, respectively, in the equatorial plane, Figure 8b. The In-O(ETS) distance is about 2.02 Å.

In the case of the mesoporous adsorbent SBA-15, the indium structure interacts with the silanol groups on the silica surface and can form one (Figure 9a) or two (Figure 9b) In–OSi bonds of approximately 2.015 Å.

Given that the structure of the SBA-15 adsorbent surface has silanol groups, three covalent bonds can form in an acidic environment between indium atoms and the oxygen atoms in the upper layer of the surface, Figure 10. In this case, the indium structure has a deformed octahedron configuration with two In–OSi bonds in the equatorial plane and one in the axial plane. In this configuration, the sulfate structure can be easily removed in the form of sulfuric acid. At the same time, when the pH of the system increases, the structure of indium in this configuration can pass into the hydroxylated form. This aspect justifies the higher adsorption capacity in the case of the SBA-15 structure compared to ETS-10. In the case of the ETS-10 structure, the formation of three In–O-type bonds with the adsorbent surface is very difficult, because in this configuration, the distance between the oxygen atoms on the surface is greater, being interspersed by Ti atoms. On the other hand, the theoretical calculation results for the structure of InSO_4_^+^ hydrated with 5 water molecules, assimilated with a sphere, indicated a diameter of approximately 8 Å. Considering the diameter of the pores in the adsorbent: 7.5 nm and 0.8 nm for SBA-15 and ETS-10, respectively [28], it can be stated that a larger amount of indium sulfate will enter and be retained in the cavity by SBA-15 compared to ETS-10.

In both cases, between the hydration water molecules and O atoms on the surface of the adsorbent structure, physical interactions with the hydrogen-bond-type are formed, which contribute to the facilitation of adsorption and the binding of the In atom to the mesoporous structure.

### 3.5. Desorption Studies

Important criteria in the selection of the sorbents for pollutants removal from wastewaters are their high stability and reusability. The application of 0.01 M HCl as an eluent was demonstrated to be an appropriate method to regenerate the adsorption abilities of both adsorbents. The values of the adsorption capacity after six sorption/desorption cycles are presented in Figure 11.

As can be seen, the adsorption capacity values of SBA-15 and ETS-10 decreased from 99.27% and 96.48% to 90.49% and 86.64% in the first cycle, and to 67.61% and 63.36% in the sixth cycle, respectively. The removal efficiency values decreased between 4% and 10% for SBA-15 from the first to the sixth cycle, and between 5% and 10% for ETS-10, respectively, and this showed that both the adsorbents could be regenerated and reused for the indium removal in multiple cycles.

## 4. Conclusions

In the present study, silica SBA-15 and titanosilicate ETS-10 materials were utilized for indium ion adsorption from aqueous solutions. The process of indium ion adsorption was quick (no more than 45 min) and pH-dependent. A maximum titanosilicate ETS-10 and silica SBA-15 removal of 99% was achieved at pH 3.0 and 6.0, respectively. An acidic pH was more favorable for metal ion recovery, while at high pH values indium removal occurred due to both adsorption and precipitation. The Langmuir model better described the experimental data, suggesting that indium adsorption on the studied sorbents was due to adsorption onto the monolayer. The maximum adsorption capacity calculated from the Langmuir model was 366 mg/g for titanosilicate ETS-10 and 2036 mg/g for silica SBA-15. The kinetics of the indium(III) adsorption agreed well with the Elovich model for silica SBA-15 and pseudo-second order model for titanosilicate ETS-10. The process of indium(III) adsorption was temperature-independent, at temperatures ranging from 20–50 °C, and metal removal was on the level of 94–99%. Thermodynamic studies showed that the removal of indium using the studied sorbents was spontaneous in nature, endothermic for silica SBA-15, and exothermic for titanosilicate ETS-10. According to theoretical simulations, three covalent bonds can be formed at an acidic pH between indium atoms and the oxygen atoms in the case of silica SBA-15, due to presence of silanol groups on the surface. At the same time, when the pH of the system increases, the structure of indium in this configuration can pass into the hydroxylated form. This higher adsorption capacity of the silica SBA-15 at higher pH values compared to titanosilicate ETS-10 can be explained by the hydroxylation of indium ions as well as the large diameter of silica SBA-15 pores. The high efficiency of indium(III) desorption and the maintenance of a high adsorption capacity allow for multiple uses of studied adsorbents.

## Figures and Tables

**Figure 1 materials-16-03201-f001:**
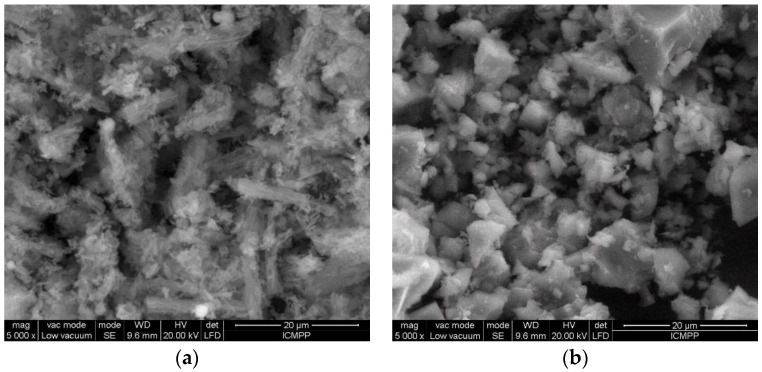
SEM microphotographs of used sorbents (**a**) silica SBA-15 and (**b**) titanosilicate ETS-10.

**Figure 2 materials-16-03201-f002:**
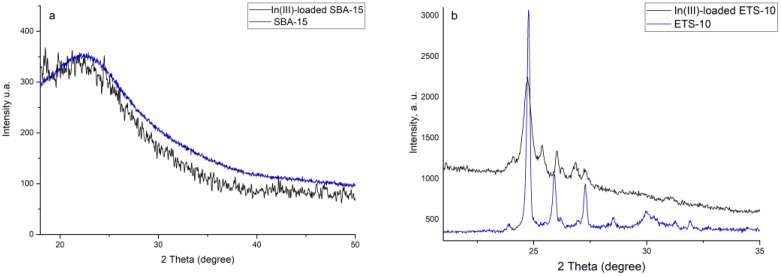
XRD patterns of (**a**) silica SBA-15 and (**b**) titanosilicate ETS-10 before and after In(III) -adsorption.

**Figure 3 materials-16-03201-f003:**
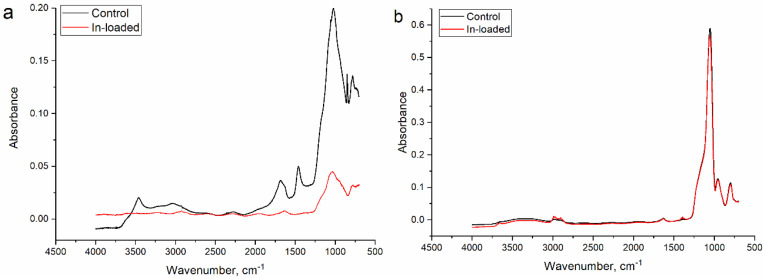
FTIR spectra of studies sorbents (**a**) silica SBA-15 and (**b**) titanosilicate ETS-10.

**Figure 4 materials-16-03201-f004:**
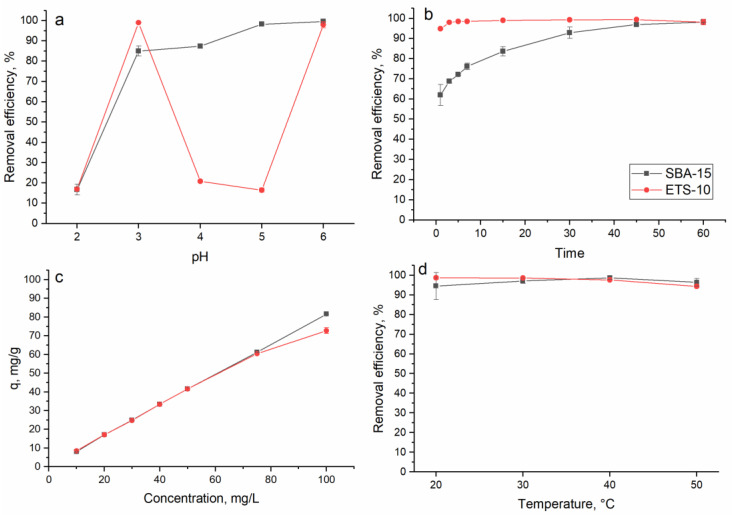
Effect of (**a**) pH (C_i,In_ 10 mg/L, time 60 min, temperature 22 °C), (**b**) time (C_i,In_ 10 mg/L, pH 3.0 for ETS-10 and 5.0 for SBA-15, temperature 22 °C), (**c**) indium concentration (pH 3.0 for ETS-10 and 5.0 for SBA-15, time 60 min, temperature 22 °C) and (**d**) temperature (C_i,In_ 10 mg/L, pH 3.0 for ETS-10 and 5.0 for SBA-15, time 60 min 22 °C) on the adsorption of indium ions on studied sorbents.

**Figure 5 materials-16-03201-f005:**
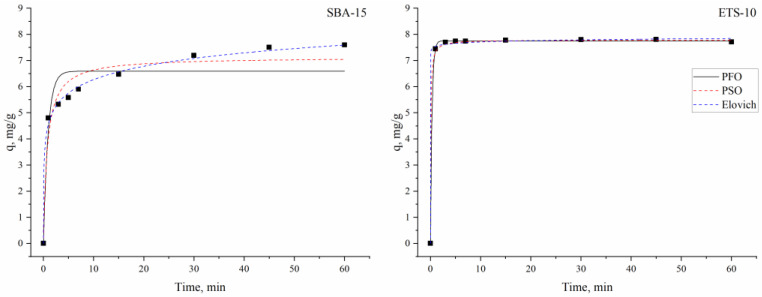
The kinetic models describing indium ion sorption on silica SBA-15 and titanosilicate ETS-10 sorbents.

**Figure 6 materials-16-03201-f006:**
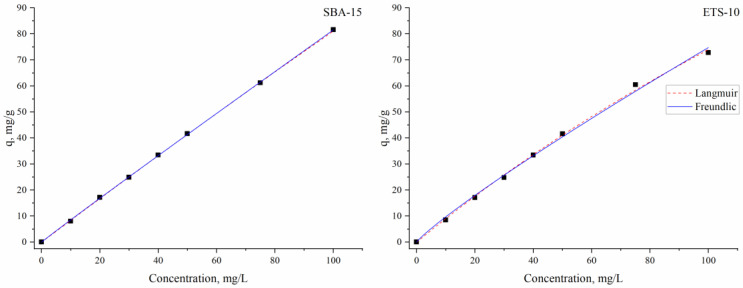
The equilibrium models describing indium ion sorption on silica SBA-15 and titanosilicate ETS-10 sorbents.

**Figure 7 materials-16-03201-f007:**
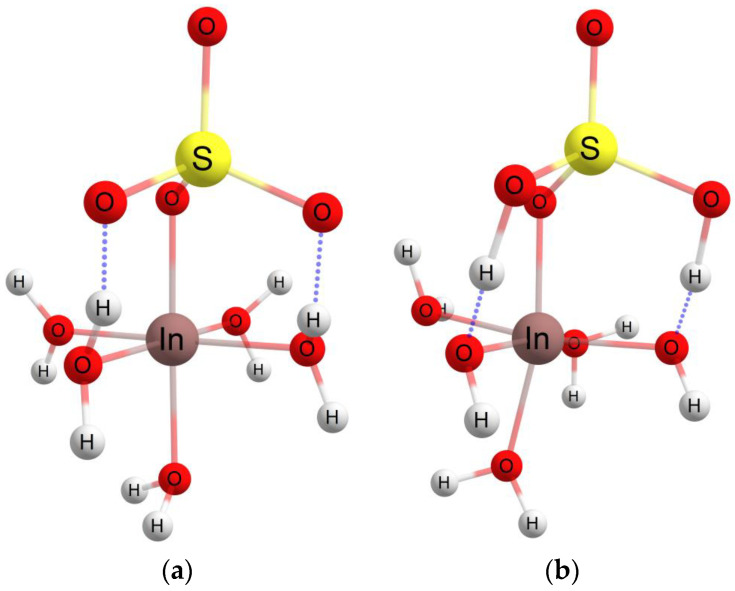
The ideal spatial structure of the In(H_2_O)_5_(SO_4_)^+^- (**a**) and In(OH)_2_(H_2_O)_3_(H_2_SO_4_)^+^- (**b**) (The blue dotted line illustrates the hydrogen bond type interaction).

**Figure 8 materials-16-03201-f008:**
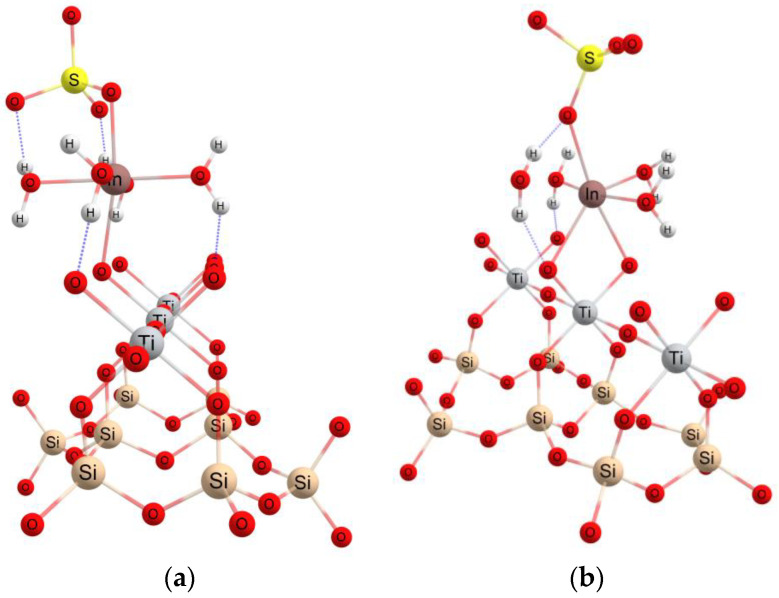
The interaction with the ETS-10 adsorbent surface and the formation of one (**a**) or two (**b**) bonds with oxygen.

**Figure 9 materials-16-03201-f009:**
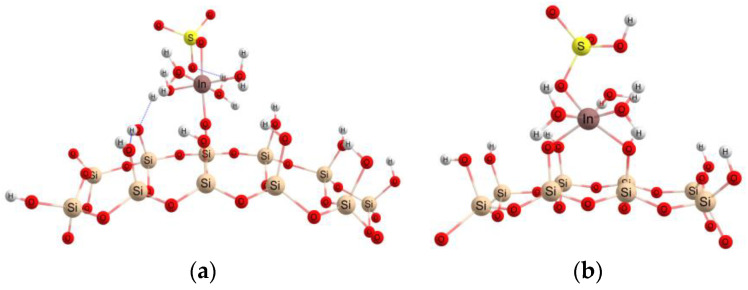
Formation of one (**a**) or two (**b**) In–OSi bonds in the case of interaction with the SBA-15 adsorbent surface.

**Figure 10 materials-16-03201-f010:**
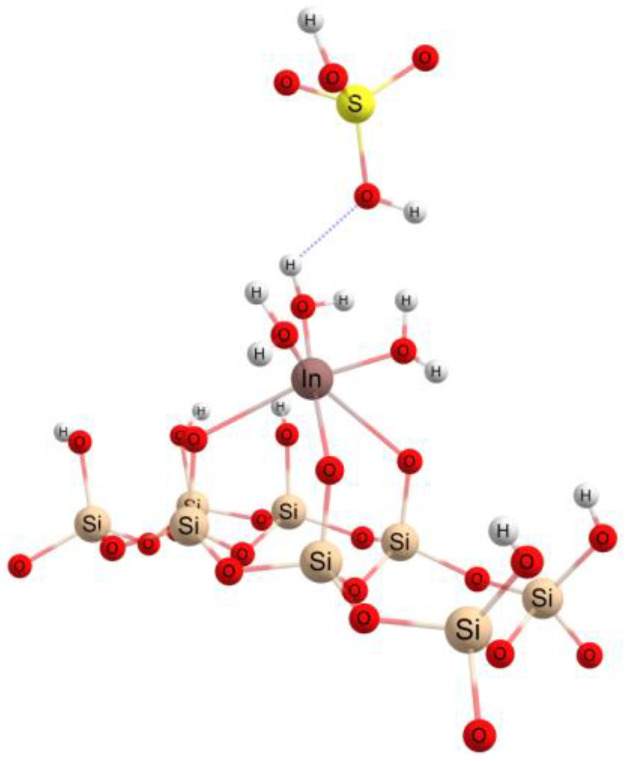
The interaction of the In structure with the SBA-15 surface and formation of three In–OSi covalent bonds.

**Figure 11 materials-16-03201-f011:**
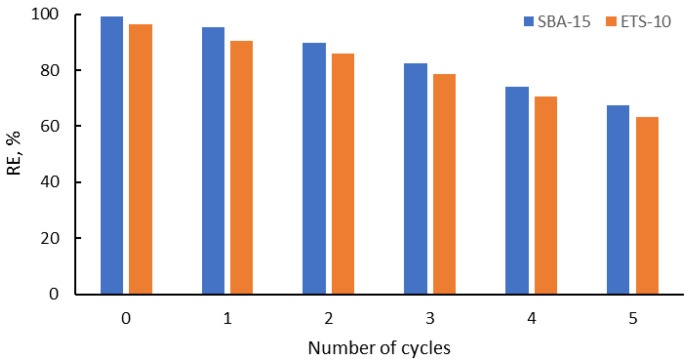
Reusability of the silica SBA-15 and titanosilicate ETS-10 in the removal of In(III).

**Table 1 materials-16-03201-t001:** Equilibrium isotherms and kinetic parameters for the sorption of indium on silica SBA-15 and titanosilicate ETS-10 sorbents.

**Isotherms**										
	**Langmuir**					**Freundlich**				
Sorbent	q_m_		b		R^2^	K_F_		n	R^2^	
SBA-15	2036 ± 593		4.15 × 10^−4^ ± 1.25 × 10^−4^		0.99	0.89 ± 0.02		1.02 ± 0.007	0.99	
ETS-10	366 ± 64.5		0.0025 ± 5.28 × 10^−4^		0.99	1.26 ± 0.17		1.3 ± 0.04	0.99	
**Kinetics**										
**Sorbent**	**Pseudo-first order**				**Pseudo-second order**			**Elovich**		
	q_cal_	q_e_	k_1_	R^2^	q_e_	k_2_	R^2^	α	Β	R^2^
SBA-15	6.7 ± 0.01	6.6 ± 0.3	1.1 ± 0.39	0.89	7.13 ± 0.28	0.19 ± 0.06	0.95	366 ± 140	1.4 ± 0.068	0.99
ETS-10	7.8 ± 0.03	7.8 ± 0.02	3.2 ± 0.13	0.99	7.8 ± 0.03	1.99 ± 0.34	0.99	3.76 × 10^−43^ ± 2.63 × 10^−4^	13.6 ± 0.93	0.99

**Table 2 materials-16-03201-t002:** The comparison of maximum sorption capacity of In(III) ions onto different adsorbents.

Adsorbent	Conditions	q, mg/g	Reference
Mesoporous silica SBA-15	pH = 6, t = 22 °C	2036	Present study
Titanosilicate ETS-10	pH = 3, t = 22 °C	366	Present study
Chitosan-coated bentonite beads	pH = 4, t = 20 °C	17.89	[22]
Raw chitin	t = 25 °C	152.75	[2]
Supercritical-modified chitin	t = 25 °C	137.69	[2]
Spent coffee grounds	pH = 4, t = 25 °C	181.8	[9]
UiO-66	pH = 3, t = 25 °C	11.75	[37]
CoFe_2_O_4_-zeolite	pH = 5, t = 25 °C	0.0947	[25]
Poly(vinylphosphonic acid-co-methacrylic acid) microbeads	pH = 6, t = 25 °C	80.37	[20]
Modified solvent impregnated resins	pH = 3, t = 25 °C	44.25	[21]
Carbon nanotubes	pH = 7, t = 20 °C	40	[23]

**Table 3 materials-16-03201-t003:** Thermodynamic parameters of indium sorption on silica SBA-15 and titanosilicate ETS-10.

Sorbent	∆G°, kJ/mol				∆H°, kJ/mol	∆S°, J/mol·K	R^2^
	293 K	303 K	313 K	323 K			
Silica SBA-15	–7.7	–8.6	–9.4	–10.3	17.3	85.2	0.93
Titanosilicate ETS-10	–11.1	–10.1	–9.2	–8.2	–38.8	–94.6	0.99

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
