# Peer review of "Adsorption Capacity of Silica SBA-15 and Titanosilicate ETS-10 toward Indium Ions"

_materials, 2023, doi:10.3390/ma16083201_

Round 1

Reviewer 1 Report

The manuscript follows the typical way of adsorption study by investigating parameters such as pH, time, temperature, concentration and fitting the experimental data with kinetics and isotherm models. Introduction must be improved by stating why SBA-15 and ETS-10 are selected as the adsorbents for indium ions. These two materials are not easy to prepare and the preparation might be costly, compared to bio-sorbent or activated carbon. I also have the following comments:

-       Details of experimental conditions in Fig. 3 are not complete, i.e., Fig. 3c was obtained at what time?, At which pH, the result in Fig. 3d was obtained?

-       On page 7, why temperature does not affect the indium removal?

-       The result in Fig 4 and the kinetics study was conducted at only the initial indium concentration of 10 mg/L?  Adsorption capacity at equilibrium (qe) determined from experiment should be reported and compared with the calculated values from pseudo first- and second-order models in Table 1.

-       How are the maximum adsorption capacities of the present study in Table 2 calculated since the values are different from those previously reported in Table 1?

-       The discussion in Section 3.4 is interesting. However, authors only discuss in terms of interaction between adsorbent and adsorbate (chemical adsorption). The physical adsorption in terms of pore size and volume as well as the specific surface area should also be discussed. Does SBA-15 give higher adsorption capacity than ETS-10 due to the much higher specific surface area (803 vs 31 m2/g)? In this case, physical characteristics is more important than chemical properties. Do the chemical properties in term of functional group is important for the adsorption of indium ions? More discussion should be added.  

Author Response

Thank you very much for your valuable comments. All improvements are in red in the revised form of the manuscript.

Best regards!

Reviewer 2 Report

The following comments need to be addressed before it can be considered for this journal.

1. What is the concentration of In and volume of In solution was used. Further the what is amount of adsorbent was used in each experiment.

2. Measure pzc values of both adsorbents and discuss the effect of pH in relation to zpc.

3. Perform BET analysis to see the pore volume, surface area of materials to better understand the adsorption ability.

4. Perform XRD analysis and EDX analysis before and after adsorption to further get insights of adsorption process.

5. Introduction need to be improved using more citations. Discuss other materials for metals, to further highlight these adsorbents. Some examples are (https://doi.org/10.1016/j.jtice.2013.09.012, https://doi.org/10.1080/19443994.2012.752765)

6. Sustainability is very important aspect. Perform regeneration study to see the stability of adsorbents.

7. In thermodynamics calculations, the Kd value calculated wrongly. The Kd must be dimension less. The equation authors used has Kd with unit. So read literature to find correct calculation of Kd dimension less. Otherwise all thermodynamics calculations are wrong.

Author Response

Thank you very much for your valuable comments. All the changes  are in red in revised form of the manuscript.

Best regards!

Reviewer 3 Report

Dear Editor

In this paper the authors have presented a valuable work on indium ions adsorption using different two types of silica materials, one mesopore (SBA-15) and one micropore (ETS-10). Although the manuscript is appropriately organized and the results have been explained well, there are some issues need to be addressed before considering for publication. So, I would say Major revision is needed for the present work.

1.       Abstract needs to be revised by adding optimal operating variables to reach maximum adsorption capacity in both samples.

2.       In abstract and conclusion sections, it should be mentioned that the maximum adsorption capacity has been reported according to Langmuir model.

3.       In introduction, the reason of the selection of those types of silica materials is not clear. Complementary explanations should be added.

4.       In introduction, a brief history and differences between two samples should be explained.

5.       The purity of each chemical and agent used during the synthesis must be added in the manuscript.

6.       The specification of analytical instruments used for material characterization must be added in the manuscript.

7.       XRD analysis should be taken for both samples.

8.       The pore sizes of ETS-10 must be reported.

9.       Why haven’t the authors mentioned any simulation results in the abstract and conclusion?

1.   Conclusion needs to be revised by adding more quantitative data.

Author Response

(The authors gave the same response as above.)

Reviewer 4 Report

The present manuscript contains interesting results, but in my opinion it is not adequate for publication in the present form.

In page 5, line 19 (please, number the lines!!!), the authors tell “The detailed characteristics of the analyzed sorbents is presented in [29]”. I suppose that there is a mistake here, and that they are actually referring to reference 28, which is a previous paper from the same group. Looking at this reference, I observe that Figures 1 and 2 were already published in this reference. Even Figure 2 seems to be the same, although one of the spectra belongs to the solids after adsorption, and the cation was different in the previous paper. If the authors have previously reported the synthesis and characterization of these solids, this must be indicated, and a very brief summary of the results may be given.

The results of In adsorption are interesting. In the introduction it is said that “Indium in nature is present associated with zinc, iron and copper in minerals and prior to using it has to be recovered from these minerals”. From these sentence, one expect that the adsorption in the presence of these other elements should be studied, as a method for concentrate this element. Have the authors explored this possibility? Should In be concentrated in this way or a preferential adsorption of the other elements should be found?

Page 2, section 2.1. The formula 3.4Na2O:1.5K2O:TiO2:5.5SiO2:150H2O is given for the titanosilicate. How was this formula calculated? Is it a theoretical formula, or an experimental one? Later, in page 4, in the discussion of SEM results, it is written “Na+/K+ ratio of 2.8 calculated for titanosilicate”, which does not agree with the previous formula. The ratio Ti/Si agrees with the targeted considering the amounts of each precursor used? And how was the amount of water calculated?

Just after the formula, it is written “NaCl NaCl”, Delete one of them.

Page 5, IR results: “The band at 3400 cm1 could be attributed to OH functional groups”. What mean the authors with “functional groups”? This band belongs to water molecules, not to hydroxyls bonded to metal elements, M-OH, which may appear at higher wavelength (a weak shoulder in SBA and a small peak in the titanosilicate, best observed after interaction with In).

In the same paragraph, “Adsorption of indium ions resulted in the decrease of the intensity of the functional groups”. A great decrease in the intensity of the peaks is observed (for SBA silica), but it seems to be due to the use of a lower amount of sample. In the case of the titanosilicate, such an effect is not observed, which is much more reasonable.

Next paragraph: “ at 1645 cm1 to OH groups”. This band does not correspond to OH group, but to the bending vibration of water molecules.

Two lines later, “Peat at 1380 cm1 can be assigned to nitrates ions”: I cannot understand this, as nitrate has not been used as precursor of any element, so this anion cannot be in the samples.

Also in this paragraph, “In the spectrum of the sorbent obtained after indium ions adsorption changes in the intensities of the peaks in the area of OH groups after adsorption”. Sincerely, I do not observe these changes. In SBA, the decrease of water is observed, but nothing can be said about hydroxyls. In the titanosilicate, even a small increase in the intensity of hydroxyl band can be observed, that may be due to the hydration of the solid during the adsorption process. But I do not observed changes that can be assigned to the coordination to the cation.

In section 3.2: “Further increase of the pH resulted in the drastic decrease of indium removal up to 16% at pH 5.0, that can be attributed to a reduction in the concentration of positively charged groups on the sorbents surface”. This is contradictory with the previous statement, when discussing the influence of pH. There, the low adsorption at low pH was assigned to the presence of protonated groups, that may cause repulsion with the cation. So, if the pH increases, adsorption should also increase.

In the same paragraph, “at pH 6.0 titanosilicate ETS-10 removal capacity toward indium increased up to 74%”. Looking at the Figure, the adsorption at this pH seems to be approximately 95%.

The conditions for the experiments in figure 3 must be indicated. I mean, when analyzing the influence of pH, the time, the temperature … should be fixed. The same when analyzing the influence of time, the other variables should be fixed. All this information is lacking.

Page 6, “High indium removal at higher pH values can be explained by its precipitation”. At what pH indium begins to precipitate?

Page 9: How can the authors explain that adsorption causes an increase of entropy for one adsorbent and a decrease for the other?

Legend of Figure 6.In the second formula, In(OH)2(H2O)3(SO4)+, the charge of this species should be negative (-1), not +1. Besides, looking at the figure, it seems that it contains five water molecules (or 3 water, 2 OH and one H2SO4), in these cases the charge is negative.

Author Response

(The authors gave the same response as above.)

Round 2

Reviewer 1 Report

The manuscript has been corrected according to the reviewers’ comments.

Author Response

Thank you very much!

Best regards!

Reviewer 2 Report

Few minor changes before it can be considered.

This comment is not addressed.

Introduction need to be improved using more citations. Discuss other materials for metals, to further highlight these adsorbents. Some examples are. Cite these referecnes. (https://doi.org/10.1016/j.jtice.2013.09.012, https://doi.org/10.1080/19443994.2012.752765)

Author Response

Thank you very much for valuable remarks.

Best regards!

Reviewer 3 Report

Dear Editor

The authors have responded to all raised comments of the respected reviewer appropriately. So, the manuscript can be accepted in its current version.

BR

Author Response

Thank you for your time spent with the evaluation of the manuscript.

Best regards!

Reviewer 4 Report

For cubic centimeters, "cm3" is more correct than "cc".

  The description of the XRD patterns  is confusing. First, what is the "control" diffractogram in these figures? In the case of titanosilicate, the authors claim both amorphous and crystalline solids, indicating that the crystalline phases are anatase and rutile. What about the titanosilicate? Was titanosilicate formed, and besides, certain amounts of anatase and rutile were also present? Please try to better explain these results.   About the desorption experiments, it is mentioned that HNO3 or HCl solutions were used. Why two different solutions? The authors should explain why it was expected that In(III) was desorbed under these conditions. After contact with the acids for 1 hour, were the solids washed, dried... before new experiments adsorption.   English style must be revised, mainly in the sentences added in red in this version.

Author Response

Thank you for valuable remarks. All the improvement are in green in the revised manuscript.

Best regards!
